# Whose research benefits more from Twitter? On Twitter-worthiness of communication research and its role in reinforcing disparities of the field

Chung-hong Chan[1]*, Jing Zeng[2], Mike S. Schäfer[3]

**1** GESIS - Leibniz-Institut für Sozialwissenschaften, Mannheim, Germany, **2** Department of Media and Culture Studies, Utrecht University, Utrecht, Netherlands, **3** Department of Communication and Media Research, University of Zurich, Zurich, Switzerland

\* chung-hong.chan@gesis.org

**Data Availability Statement:** All data and files are available from OSF (doi: 10.17605/OSF.IO/YCX6J).

**Funding:** The author(s) received no specific funding for this work.

## Abstract

Twitter has become an important promotional tool for scholarly work, but individual academic publications have varied degrees of visibility on the platform. We explain this variation through the concept of *Twitter-worthiness*: factors making certain academic publications more likely to be visible on Twitter. Using publications from communication studies as our analytical case, we conduct statistical analyses of 32187 articles spanning 82 journals. Findings show that publications from G12 countries, covering social media topics and published open access tend to be mentioned more on Twitter. Similar to prior studies, this study demonstrates that Twitter mentions are associated with peer citations. Nevertheless, Twitter also has the potential to reinforce pre-existing disparities between communication research communities, especially between researchers from developed and less-developed regions. Open access, however, does not reinforce such disparities.

## Introduction

Social media has permeated almost every aspect of society. This includes the realm of scholarly communication, where digital platforms have become important tools for informational, networking, and promotional purposes [1]. Twitter in particular is widely used to disseminate academic publications in order to amplify their visibility and, by extension, further their uptake and boost their citations [2–4]. In turn, social media engagement, especially Twitter performance, is also used to measure author and publication impacts as altmetrics [5–7].

Despite the increasingly common practice of utilizing social media for scholarly purposes, the degree of benefits researchers receive vary significantly [8]. New communication technologies advantage certain scholars over others, and inequities persist, challenging the optimistic view that widely available and popular social media democratizes scholarly communication [9, 10]. In keeping with prior studies of the "rich-gets-richer effect" in scholarly communication, i.e. researchers from privileged regions or institutions benefit more from it than the

**Competing interests:** The authors have declared that no competing interests exist.

underprivileged [11], this paper introduces the concert of *Twitter-worthiness* and examines its implications. We define Twitter-worthiness, in the context of the current study, as factors making certain academic publications more likely to be tweeted than others. As the first research question, this study asks: *which factors contribute to a publication's Twitter-worthiness (RQ1)?* The second question this study investigates is: *how is visibility on Twitter associated with citations received by a paper (RQ2)?* Additionally, this study investigates: *how do the factors predicting Twitter visibility and citations manifest differently according to the geographical locations (RQ3)?*

To examine the factors that make scholarly work more likely to be tweeted and how it is associated with citations, statistical analyses were conducted of 32187 studies published in communication journals. These research inquiries are not only highly relevant to but also much needed in the field of communication studies, wherein disparities in visibility and impacts between its research communities are pervasive [11, 12]. In recent years, efforts have been made by the academy members [13] and leadership [14] to recognise and tackle issues related to representation and equitable participation. In prior studies, these issues have been critically reflected upon, both on the structural level and in the context of research culture [13, 15, 16]. Findings from the current study contributes to the discussion by illuminating how social media may further exacerbate disparities between privileged and underprivileged research communities.

While drawing attention to questions related to power and equity in the field of communication studies and to Twitter, this study also provides methodological instruments that can be applied to different disciplines, or to other social media especially in regions where Twitter's role in scholarly communication is not prominent. Issues discussed in the current paper are not necessarily unique to the field of communication studies, and should be examined and discussed in other disciplines.

## Science communication & digital media

Research on the relationship between science and the public has a long tradition (see [17] for an overview). Prior studies have mostly focused on science's relation to news media, and demonstrated that science has established itself as an important topic in legacy media, and that academics are regular participants of media debates on topics ranging from politics to public health [18, 19]. They have also shown that both academics and educational institutions rely on mainstream media to reach the general public as well as to boost their impact [20]. Many have argued that both academics and institutions are now under increasing pressure to achieve public visibility [21].

The proliferation of digital media has changed the interplay between science and the public. Social media has been adopted quickly, albeit unevenly, by scholars [3] and it has become increasingly common for researchers to use social media for public communication [3, 22]. Scholars of science communication have discussed the potential of social media, meandering between optimistic and pessimistic accounts, for opening up educational resources [23], facilitating discussion on scientific topics between scientists and citizens [24], enhancing the social impacts of research [25], and facilitating networking among researchers [26]. While some emphasize the democratizing potential of new communication technologies to foster a more equal and inclusive context for accessing and engaging with science, others present a more critical viewpoint on digital media affordance, concerning issues related to efficacy and equity [10, 27].

Early on, academics' and universities' adaptation of new communication technologies was commonly discussed in relation to its ability to create room for engagement and to bridge the

gap between academic communities and the general public [9]. However, empirical findings show that individual researchers, project groups, and institutions often use Twitter for advertising publications or other research updates [28], similarly to the way "other commercial, political, or societal actors do in their marketing and PR efforts via Twitter" [3]. Veletsianos [4], for example, categorized tweets from 100 researchers qualitatively and showed that promoting publications is one of the key features of academics' usage of Twitter. A large-scale survey of academics across disciplines in 2014 came to similar results [1].

However, not all scholarly publications are equally popular on social media. In spite of their decentralized and open architecture, social media often does not present a fully equitable environment and the power disparity between elite and non-elite users remains prominent. Social capital, expertise, and other credentials are required to become influential in the seemingly inclusive environment [29], and in the context of science communication, social media may exacerbate the disparity in visibility and impact that exists between academic output from different disciplines, geographical locations, and individual researchers [9, 11, 30]. Scholarship explains the differing visibility of publications from communication studies by historical factors, such as the influence of colonialism and the early establishment and institutionalization of communication studies in Europe and the US [31, 32]; geo-economic factors, such as lack of resources [31]; and linguistic factors, such as the dominance of English [12, 33].

In this study, we are interested in how social media deployment, with Twitter as an example, may advantage certain academic output or individual scholars more than others.

## Factors determining Twitter-worthiness

We assume that the visibility of publications from communication on Twitter can be captured with the concept of Twitter-worthiness. It conceptualizes the characteristics that make certain content, in our case academic publications, more likely to be shared on Twitter.

Twitter-worthiness is inspired by news-worthiness, an established concept from journalism studies. Rooted in news value theory, news-worthiness assumes that events or topics have characteristics that increase, or decrease, their chances of being taken up for news media coverage [34]. For decades, scholars aimed to identify factors explaining journalists' decision about which topics or events are worthy enough to report on, and to present most prominently [35–37]. Established frameworks of news values identify factors like timeliness, relevance, conflict, entertainment, surprise or the involvement of power elites, and argue that the more of these factors are represented in a topic or event, the more like it is to be covered in the news media [38].

Similarly, we investigate factors that are likely to make publications from communication studies more visible on Twitter, describing these factors as Twitter-worthiness (RQ1). Following the tradition of news-worthiness research, our conceptualization and measure of Twitter-worthiness focuses on the topical, author-based and geographical attributes of scholarly outputs:

1. *Hot research topics*: We assume that the research topic of a publication impacts its visibility on social media. Previous studies have shown communication researchers are more likely to write about certain topics [39] and cite papers from popular [40, 41] or "orthodox" topics [11]. In all of the cited studies, papers researching social media are more likely to be written and cited. As the scholarly community represents the vast majority of users sharing links to scientific publications, *we expect papers researching social media to be more Twitter-worthy than others (H1)*.

2. *Geographical origin*: Regional difference is another dimension of *Twitter-worthiness*. Prior studies have shown ample evidence indicating the dominance of the Global North in communication studies [11, 12, 30]. Lauf's [12] study of SSCI-indexed communication journals reveals that 1) two out of three articles were from the US; and 2) the US and other English-speaking countries together contributed to over 85 percent of publication in the field. Demeter [11] shows a significant correlation between citations and GDP of a country, and a predominance of North America and Western Europe in the field, with over 80 percent of authors from the region. Contributing to this regional dominance, according to Demeter [11], are not merely the economic strength of North America and Western Europe, but also their "received history" of the field. The discipline's origin in the US, as well as the earliest establishment of university-level communication studies in the US and Western Europe [32], is also important historical factors contributing to these regions' contemporary dominance. While most existing studies investigating regional dominance of communication studies focus on journals, our study empirically assesses how communication studies' visibility on Twitter is affected by their geographical origin. We hypothesize that *papers from certain countries, e.g. from the U.S. and the Global North, are more Twitter-worthy than others (H2).*

3. *High impact journals*: The place of publication, i.e. the journal, also plays a role in amplifying the visibility and dissemination of scientific output [11]. Callaham et al. [42] show that the impact factor of the publishing journal is the most important factor predicting the dissemination of papers submitted to the Society for Academic Emergency Medicine meeting. Likewise, Wang et al. [43] analyze papers published in astronomy and astrophysics journals and show that the reputation of a journal has a great influence on a paper's citation trend in future. Instead of focusing on academic citation, we test how the average scholarly impact of a journal influences its publication's visibility on social media. We hypothesize that *papers published in high impact journals are more Twitter-worthy than other papers (H3).*

4. *Open access*: Another feature of a publication that may relate to Twitter-worthiness is whether it is published open access (OA), i.e. freely available to all audiences. Scholarly communication in OA journals has grown considerably in past decades [44, 45]. For most disciplines, prior research has demonstrated a positive impact of OA on the academic impacts of publications. For example, across philosophy, political science, electrical and electronic engineering and mathematics, Antelman [46] finds that articles published in OA journals have greater impacts than other papers. Zhang's [47] study of Web citation of research published in *Journal of Communication* and *New Media & Society* shows that publications with OA have twice as many citations. So far, the impact of OA has mostly been assessed towards academic citations, not towards social media presence. The few available studies that did analyze this connection showed a clear impact. For instance, one study of 1761 publications in *Nature Communications* shows that OA articles have more social media presence than non-OA articles [48]. Accordingly, we hypothesize that *OA papers are more Twitter-worthy than other papers (H4).*

## Is visibility on Twitter worthy or worthless? Effects on peer citations

A considerable body of scholarship has tested the nexus between scholarly papers' visibility on Twitter and their citation counts. The respective results, however, remain inconclusive. Eysenbach's study [49] of 4208 tweets citing articles from the *Journal of Medical Internet Research* reveals a strong association between Twitter mentions and citations during the early days after

publication, and concludes that Tweets can be an early predictor of highly cited articles. Similar strong correlation between Tweets and early citation is also detected in Shuai et al's study [50] of preprints from *arxiv.org*.

Despite earlier evidence suggesting that mentions in tweets can be an early indicator of citations, more recent studies have shown that this correlation is weak and can be negative in certain fields, and a case in point is de Winter's [51] study of over 27,000 *PLOS One* publications. Haustein et al.'s research [52] of papers from *Web of Science* concludes that retweets and citation rates do not correlate. Similar results are discussed in other papers [6, 53, 54]. Because the impact of Twitter on citation varies significantly depending on the discipline and the journal [9], the conflicting results discussed above are not surprising.

To examine how a paper's presence on Twitter is associated with the number of citations it received (RQ2), we focus on publications of communication research, and aim to generate contextualized knowledge of how Twitter can impact outputs from this field. Studying the potential impact of Twitter on citation is not straightforward because the factors that have been hypothesized to be Twitter-worthy can also boost citations. *We assume that after adjusting for other factors, papers being tweeted more frequently also have more citations (H5).*

As mentioned in the previous section, the Global North dominates communication studies [11, 12, 30]. The relationships proposed in H1, H3, H4 and H5 might manifest differently based on the geographical locations. Thus, we investigate the interaction effects between geographical locations and the aforementioned factors (Hot research topic, high impact journals, and OA) on Twitter mentions and citations (RQ3).

## Materials and methods

### Data collection & operationalizations of variables

The basic sample of our analysis consisted of all journals indexed in the "communication" category of the 2017 edition of Clarivate's *Web of Science* (WoS)—one of the most comprehensive databases of scholarly publications available, containing more than 90 million publications over more than 100 years and across all scientific disciplines. The category includes 84 journals. During initial screening, two journals (*International Journal of Communication* and *Technical Communication*) were excluded because they did not provide digital object identifiers (DOI) for their articles and/or contained no abstracts and/or did not mention corresponding authors. The remaining 82 journals were included in the analysis.

Data collection was executed in June 2019. From the aforementioned 82 communication journals, using the time frame between 2007 and 2018, we identified 32187 journal articles on WoS. The metadata of these papers—including citation count, abstract, keywords, title, first author, and the institution of the reprint author—were downloaded as bibliographic information files and then processed in R with the *bibliometrix* package [55]. The year 2007 was chosen as the starting point for data collection, because Twitter was launched to the public in mid-2007.

We operationalized six variables relevant to our analysis:

1. *Twitter mentions* (*μ*) were operationalized as the total number of mentions a paper received on Twitter, regardless of the context or the evaluative tone of these mentions. Twitter mentions were counted using *Altmetric* data, which aims to quantify the impact of scholarly papers apart from peer citations including social media and Twitter mentions [56]. *Altmetric*, according to its website, "tracks Twitter attention in real-time via an API . . . collect[s] tweets, retweets, and quoted tweets that contain a direct link to a scholarly output." [57] Employing the Altmetric API and the R package *rAltmetric* [58], we extracted the mention counts of all

32187 articles in June 2019, with the DOIs. Let $\mu_{ij}$ denote the Twitter mentions of the $i$-th paper in the $j$-th journal.

2. The *geographical origin* of a paper (US, G11) was based on the address of a paper's corresponding author, operationalized as the "reprint author" provided by the WoS, i.e. the author one should contact in order to obtain a reprint of the paper. The geographical origin of a paper was classified into three groups: US, non-US G12 countries (hereafter G11: Australia, Belgium, Canada, France, Germany, Italy, Japan, Netherlands, Spain, Sweden, Switzerland and the United Kingdom), and non-G12 countries. This trichotomy of countries represents the current understanding of the field: The US represents the largest research community, the country in which most journals and researchers are based, and still the dominant country in terms of academic output and influence. The G11 countries are mostly Western countries (except Japan) and institutions in these countries are more likely to extend the so-called "Western-focused" canon of communication studies. Non-G12 countries are mostly from the Global South countries which are either traditionally underrepresented in communication research or lack resources to support academic research. At the same time, industrialized countries also provided some of the largest user communities during the early days of Twitter. Users from rich Anglophone and European countries were the majority. In 2012, roughly 30% of Twitter users were from the US [59]. Therefore, the current operationalization of geographical origin factors in both the geographical disparities in the research community in communication science and the users composition during Twitter's early days. Let $x_{G11,\,i}$ and $x_{US,\,i}$ denote the geographical origin of the $i$-th paper.

3. *Hot research topics* ($\theta_t$) are topics of communication research that can attract more peer citations, and are high in supply and high in disciplinary prestige—such as social media research [40]. Using a topic modeling technique [39–41], we extracted topics from the abstracts of the analyzed research papers. As we intended to extract the same "social media research" topic as Chan & Grill [40], we used a semi-supervised approach with the keyword assisted topic model (keyATM) [60].

The keyATM is an extension to the latent dirichlet allocation (LDA) [61], a traditional unsupervised topic modeling technique for modeling topic clusters in a corpus. The keyATM aims to improve the interpretability of unsupervised LDA by incorporating prior knowledge about the possible topics in the corpus with a keyword dictionary and using it to guide topic extraction. While the output of keyATM is the same as LDA (a vector of topic membership probability $\theta_t$ for each document), the keyATM requires the tokenized corpus and an additional keyword dictionary as the input. For more information about the method, please refer to either the original paper [60] or the associated R package.

For the current analysis, we tokenized the corpus of abstracts using the R package quanteda. For the keyword dictionary, the words from the "social media research" topic (i.e. *twitter, facebook, sns, tweet, blogging*) in Chan & Grill [40] were converted into keywords (*twitter, facebook, sns, tweet, blog*) and used to train a 40-topic keyATM. This ensures that our keyATM contains at least one topic that matches the existing social media topic found in Chan & Grill [40]. We used the $\theta_t$ of this topic from the keyATM as an indicator of whether a paper belongs to this topic. Let $x_{\theta_t,i}$ denote the $\theta_t$ of the $i$-th paper.

4. *High impact journals* (Q1): The 2017 edition of WoS's Journal Citation Reports provides the average number of citations per citable item published in a journal. These "Journal Impact Factors" are expressed either as numerical values or quartile values relative to other journals in the same field. In 2017, 21 journals were classified as Q1 (*Communication Monographs, Communication & Sport, Comunicar, Communication Research, Communication Theory, Human Communication Research, Information Communication & Society, International Journal of*

*Press-Politics, International Journal of Advertising, Journal of Advertising Research, Journal of Advertising, Journal of Computer-Mediated Communication, Journal of Communication, Journalism, Media Psychology, New Media & Society, Political Communication, Public Opinion Quarterly, Public Understanding of Science, Science Communication, and Telecommunications Policy*), i.e. as high impact journals. On average, publications in these journals receive a higher number of citations than publications in other journals.

In this study, we chose to use whether a journal is Q1 as the measure of its impact instead of "impact factor". The reason is that this study covers publication across a timespan over ten years, and the quartile value of specific journals is relatively more stable than the numerical "impact factor" value. Let $x_{Q1, ij}$ denote the Q1 status of the $j$-th journal publishing the $i$-th paper.

5. A publication was categorized as an *OA publication* if at least one version of the publication was available to freely access. We used Unpaywall—an open dataset of free access scholarly work—to check each paper's OA status. Data was automatically retrieved in May 2020 from *Unpaywall*, which harvests over 5,000 journal websites, OA repositories and university OA ePrint archives to determine whether publications are available for OA. We employed the Unpaywall API, using the DOIs of all included papers. Let $x_{OA, i}$ denote the OA status of the $i$-th paper.

6. The *number of peer citations* ($v$) was included based on the total number WoS citations of a given paper. It might or might not reflect the scientific quality and rigor of a paper, because there are many social factors associated with citation behaviors, such as number of coauthors of a paper [62]. Let $v_{ij}$ denote the peer citations of the $i$-th paper in the $j$-th journal.

## Statistical analysis

We employed two kinds of statistical analysis. First, we assessed descriptively how many Twitter mentions the included publications received, and how these numbers varied across the factors that we proposed to assess Twitter-worthiness, such as geographical origin, journal or OA status.

Second, we employed multilevel analysis. At this level, we considered all factors together to determine which of them are independent predictors of Twitter mentions (H1-H4) and citations (H5). As both mentions and citations are count data, a count-based, Bayesian zero-inflated mixed-effect negative binomial regression model was used. This approach was used previously [40] to adjust for the variation between journals, i.e. a varying intercept based on journals is added. Bayesian model, implemented in the R package *brms* [63], was used because our data set is nonstochastic, i.e. not involving random sampling [64]. In all multilevel analysis, we modeled count data with the age of the paper ($t$)—the amount of time since its publication —as the offset value. In effect, the dependent variables are rate of being tweeted and citation rate. For our RQ1, we add interaction terms between factors determining Twitter-worthiness and region to all models.

There might be individual differences in $\mu$ and $v$ according to characteristics of journals, e.g. countries of origin [65]. By adding a varying intercept according to the $j$-th journal as $u_{oj}$ to model the individual differences in $\mu$ and $v$, our two models are expressed as:

$$
\begin{aligned}
\log \frac{\mu_{ij}}{t_i} = {} & \beta_0 + (u_{oj}J_i) + \beta_1 x_{OA,i} + \beta_2 x_{G11,i} + \beta_3 x_{US,i} + \beta_4 x_{Q1,ij} + \beta_5 x_{\theta_t,i} \\
& + \beta_6 x_{OA,i} \times x_{G11,i} + \beta_7 x_{OA,i} \times x_{US,i} + \beta_8 x_{Q1,ij} \times x_{G11,i} \\
& + \beta_9 x_{Q1,ij} \times x_{US,i} + \beta_{10} x_{\theta_t,i} \times x_{G11,i} + \beta_{11} x_{\theta_t,i} \times x_{US,i} + \epsilon_{ij}
\end{aligned}
\tag{1}
$$

$$\log \frac{v_{ij}}{t_i} = \beta_0 + (u_{oj}J_i) + \beta_1 x_{OA,i} + \beta_2 x_{G11,i} + \beta_3 x_{US,i} + \beta_4 \mu_{ij} + \beta_5 x_{Q1,ij} + \beta_6 x_{\theta_t,i}$$
$$+ \beta_7 x_{OA,i} \times x_{G11,i} + \beta_8 x_{OA,i} \times x_{US,i} + \beta_9 \mu_{ij} \times x_{G11,i} + \beta_{10}\mu_{ij} \times x_{US,i} + \beta_{11}x_{Q1,ij} \times x_{G11,i}$$
$$+ \beta_{12}x_{Q1,ij} \times x_{US,i} + \beta_{13}x_{\theta_t,i} \times x_{G11,i} + \beta_{14}x_{\theta_t,i} \times x_{US,i} + \epsilon_{ij}$$

(2)

## Results

### Communication studies on Twitter

More than half (53.7%) of the included 30074 papers have no Twitter mention, as shown in the summary statistics presented in Table 1. The median number of Twitter mentions is 0. In sum, 115337 Twitter mentions are observed, 89% of which were received by the top 20% of the most mentioned papers. The most mentioned paper [66] received 2188 Twitter mentions, accounting for 1.9% of all mentions.

General characteristics mask stark differences across the different factors hypothesized to determine Twitter-worthiness. Table 1 shows the distribution of papers with respect to geographical origin, Q1, OA, $\mu$ and $v$. US authors contributed 46% of all included communication papers. US papers have in general a higher number of citations and Twitter mentions. However, papers from G11 countries have a much higher proportion of OA than those from the US and non-G12 countries.

As three is the upper limit of the interquartile range for $\mu$ of all publications, we use it in the following two figures to define papers with high $\mu$. Fig 1 shows the percentage of high $\mu$ papers by journal. The top 5 most visible journals on Twitter are *Political Communication, Mobile Media & Communication, New Media & Society, Journal of Computer-mediated Communication* and *Information, Community & Society*. There are 3 journals with no paper having high Twitter mentions: *Tijdschrift voor Communicatiewetenschap, Journal of African Media Studies*, and *International Journal of Mobile Communication*.

Fig 2 displays the percentage of papers with high Twitter mentions by country. In this figure, countries with at least 100 papers are included. All other countries with fewer than 100 papers are grouped under "other countries". Countries with a high level of Twitter mentions are mostly West European and North American countries. The bottom two are both Chinese-speaking regions.

### What factors determine Twitter mentions?

The regression coefficients of the multilevel Bayesian model 1 are presented in Table 2. This model supports our H1 to H4: topic, region, Q1 journals, and OA are significant independent

**Table 1. Characteristics of all included papers by region.**

|  | G11 | Non-G12 | US | All |
|---:|---|---|---|---|
| n | 9115 | 6891 | 14068 | 30074 |
| Q1 | 3092 (33.92) | 2051 (29.76) | 4567 (32.46) | 9710 (32.29) |
| OA | 1802 (19.77) | 645 (9.36) | 1525 (10.84) | 3972 (13.21) |
| $v$ | 4 (1, 12) | 3 (1, 9) | 6 (2, 15) | 5 (1, 13) |
| $\mu$ | 0 (0, 4) | 0 (0, 2) | 0 (0, 3) | 0 (0, 3) |
| $\theta_t$ | 0 (0, 0.001) | 0 (0, 0.001) | 0 (0, 0.001) | 0 (0, 0.001) |

Data are presented as either total number (percentage) or median (interquartile range). n: Total number; OA: Open access; $v$: Number of citations; $\mu$: Number of Twitter mentions; $\theta_t$: Social media topic

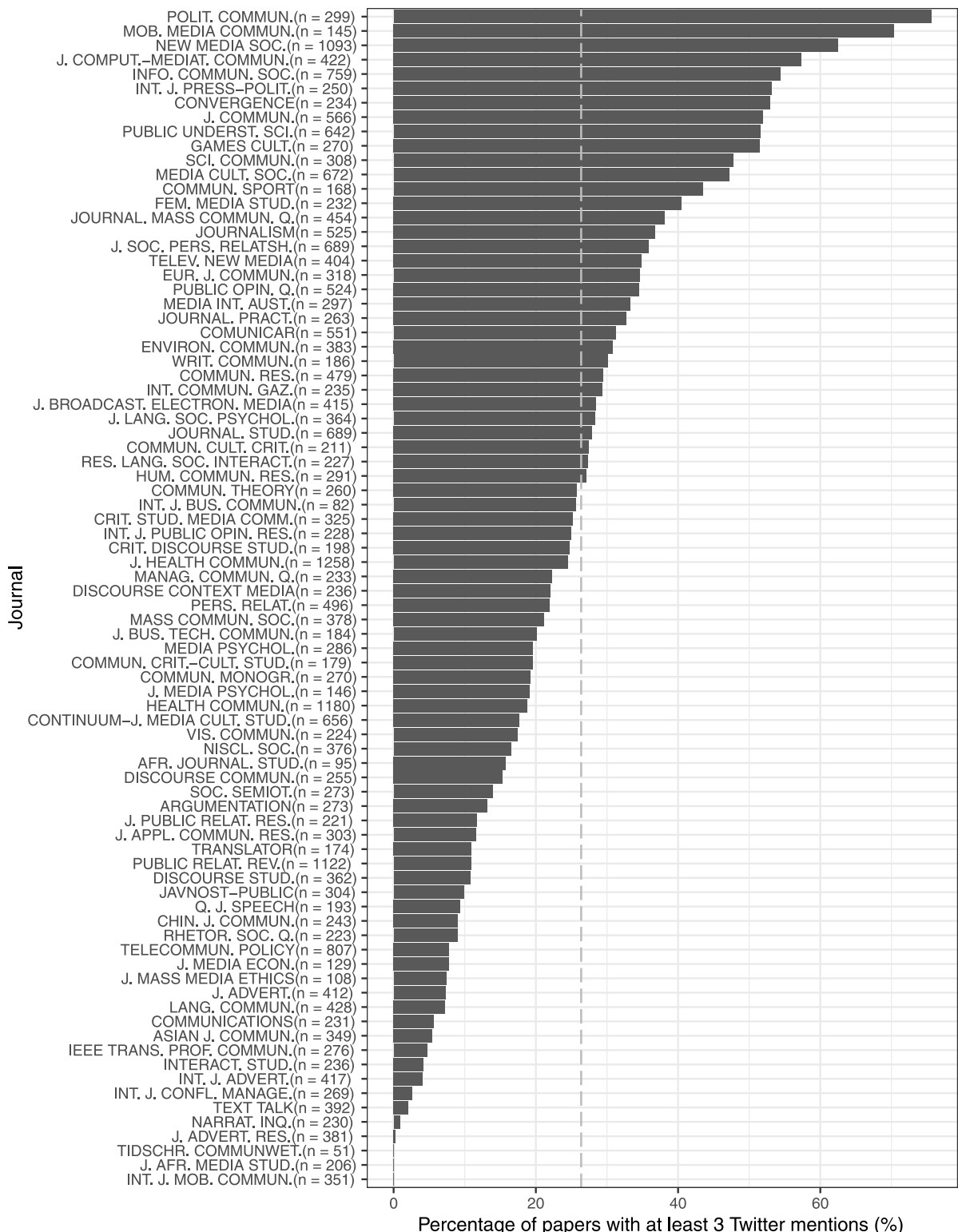

**Fig 1. Percentage of papers with at least 3 Twitter mentions by journal.** The dotted line denotes the average percentage of papers with at least 3 Twitter mentions. n denotes number of articles.

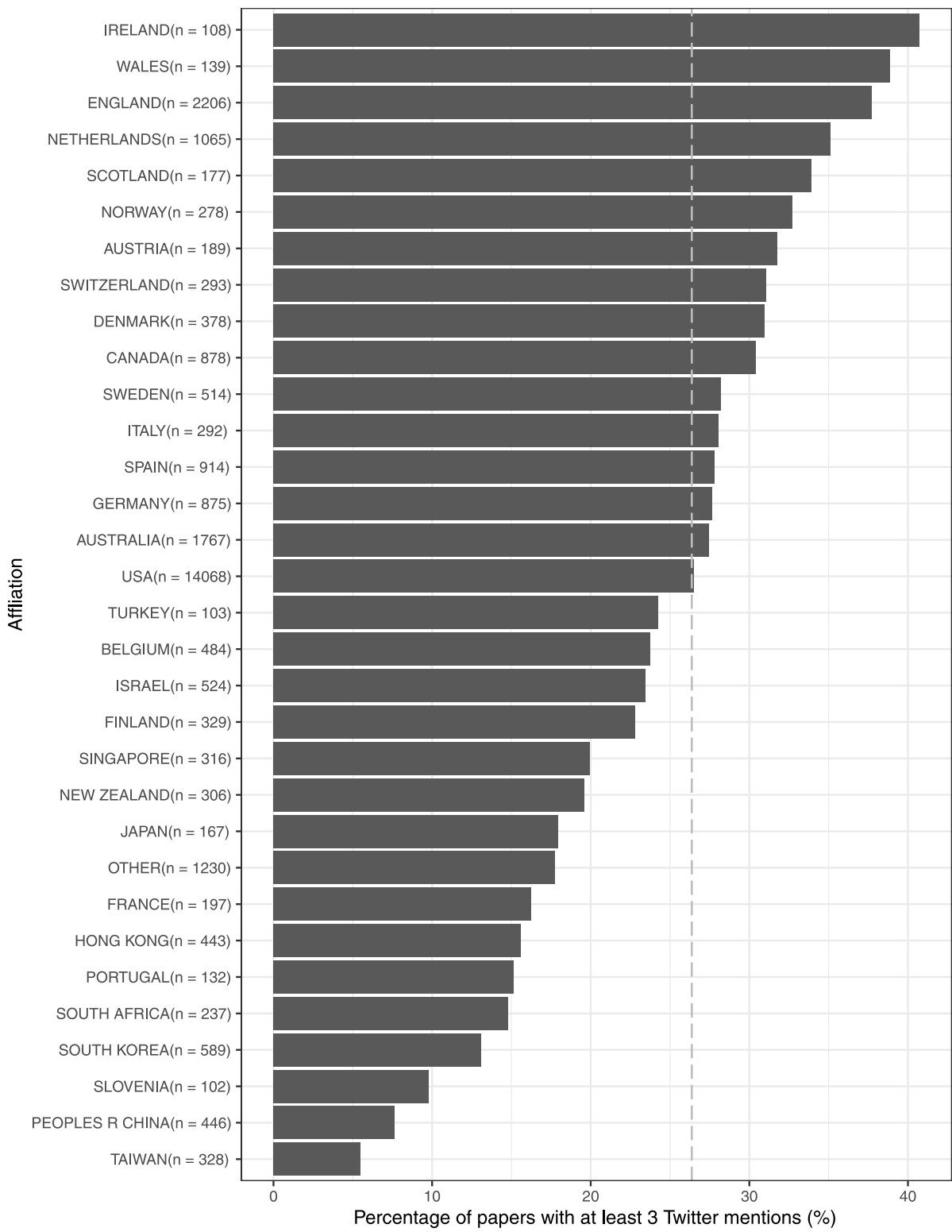

**Fig 2. Percentage of papers with at least 3 Twitter mentions by affiliation.** The dotted line denotes the average percentage of papers with at least 3 Twitter mentions. n denotes number of articles.

**Table 2. Bayesian multilevel zero-inflated negative binomial regression: Twitter mentions.**

|  | $\beta$ | 95% HDI | $e^{\beta}$ |
|---|---|---|---|
| Intercept | -1.120 | -1.531 to -0.731 | 0.326 |
| OA | 0.974 | 0.776 to 1.18 | 2.649 |
| G11 | 0.553 | 0.441 to 0.667 | 1.738 |
| US | 0.302 | 0.199 to 0.411 | 1.353 |
| Q1 | 0.690 | 0.002 to 1.43 | 1.994 |
| $\theta_t$ | 1.278 | 0.965 to 1.602 | 3.589 |
| $OA \times G11$ | -0.130 | -0.366 to 0.105 | 0.878 |
| $OA \times US$ | -0.335 | -0.57 to -0.101 | 0.715 |
| $Q1 \times G11$ | -0.009 | -0.188 to 0.167 | 0.991 |
| $Q1 \times US$ | 0.423 | 0.255 to 0.593 | 1.527 |
| $\theta_t \times G11$ | -0.099 | -0.51 to 0.315 | 0.906 |
| $\theta_t \times US$ | -0.080 | -0.482 to 0.318 | 0.923 |

Age of paper was entered as offset value; a varying intercept according to journals was entered. OA: Open access; $\theta_t$: Social media topic

predictors of Twitter mentions. However, the magnitudes of the association are different. The regression coefficient is the largest for $\theta_t$ ("social media research" topic), with the regression coefficient being 1.278. This number can be interpreted as the added number of expected log Twitter mentions in the first year. Thus, a paper on "social media research" topic has an average additional $e^{1.278} = 3.589$ Twitter mentions in the first year than papers on other topics (H1). For papers from the US and G11 countries, the expected additional Twitter mentions are 1.353 and 1.738 respectively (H2). A paper published in a Q1 journal has an expected additional Twitter mentions of 1.99 (H3). A paper published as OA has an expected additional Twitter mentions of 2.649.

When interpreting the interaction terms in Table 2, the relationships between these four factors are additional in the combination of US papers in Q1 journals (a further addition of 1.527 mentions) but antagonistic in the combination of OA US papers (a reduction of 0.715 mentions).

### Is visibility on Twitter associated with the scholarly impact of publications from communication studies?

The regression coefficients of the multilevel Bayesian model 2 are presented in Table 3. All four factors that predict Twitter mentions (OA, Country, Q1, and "social media research" topic) are also predictors of citations. This analysis supports our H5 that Twitter mentions can predict citations with an anticipated additional citation per one Twitter mention being 1.021. The interaction terms for it with G11 and US are both slightly negative.

### Discussion

This study echoes prior research that underlines social media's potential to catalyze, and amplify the influence of communication research papers [67–69]. For instance, our study confirms the positive association between Twitter mentions and citations for all communication researchers (H5).

However, our analysis of Twitter-worthiness also reveals the disparity in papers' Twitter mentions. Similar to earlier literature on news-worthiness, which has shown that the perceived

**Table 3. Bayesian multilevel zero-inflated negative binomial regression: Citations.**

| | $\beta$ | 95% HDI | $e^{\beta}$ |
|---|---|---|---|
| Intercept | -0.035 | -0.068 to 0 | 0.966 |
| OA | 0.309 | 0.216 to 0.4 | 1.362 |
| G11 | 0.183 | 0.137 to 0.228 | 1.201 |
| US | 0.326 | 0.286 to 0.368 | 1.385 |
| $\mu$ | 0.021 | 0.017 to 0.026 | 1.021 |
| Q1 | 0.575 | 0.519 to 0.632 | 1.777 |
| $\theta_t$ | 0.794 | 0.656 to 0.932 | 2.212 |
| $OA \times G11$ | -0.066 | -0.173 to 0.041 | 0.936 |
| $OA \times US$ | 0.042 | -0.068 to 0.151 | 1.043 |
| $\mu \times G11$ | -0.008 | -0.013 to -0.004 | 0.992 |
| $\mu \times US$ | -0.008 | -0.013 to -0.004 | 0.992 |
| $Q1 \times G11$ | 0.078 | 0.003 to 0.152 | 1.081 |
| $Q1 \times US$ | 0.029 | -0.039 to 0.095 | 1.029 |
| $\theta_t \times G11$ | 0.046 | -0.14 to 0.23 | 1.047 |
| $\theta_t \times US$ | 0.225 | 0.051 to 0.393 | 1.252 |

Age of paper was entered as offset value; a varying intercept according to journals was entered. OA: Open access; $\theta_t$: Social media topic; $\mu$: Number of Twitter mentions.

news-worthiness raises questions about media biases [70], our analysis of Twitter-worthiness also shows that not every researcher or every publication benefits equally from Twitter. Our statistical analysis results show that publications on "social media research" topic, from G12 countries, in Q1 journals, and with OA status are more likely to be mentioned on Twitter, or in other words: are perceived to be more Twitter-worthy.

Regardless of other factors, papers from US-based authors have a higher Twitter visibility (Table 2) and more citations (Table 3) on average. This relationship is concurrent with US papers having a higher likelihood of landing in a Q1 journal and publishing as OA than non-G12 authors (Table 1). These two factors alone (Q1 and OA) are both predictive of Twitter mentions.

Some of these relationships catalyze each other. For example, when papers from US-based authors are also in a Q1 journal, it is associated with even more Twitter mentions. Although some of them weaken the main effect, the effect is still very strong. Considering citations as the outcome, when combining the factor of a US-based author ($\beta_3 = 0.326$) and good Twitter visibility ($\beta_4 = 0.021$), it yields an antagonistic interaction effect ($\beta_8 = -0.008$). Nevertheless, papers from US-based authors with a large number of Twitter mentions still have an advantage in receiving more citations ($\beta_3 + (\beta_4 - \beta_8) \times \mu$) = 0.326 + (0.021 − 0.008) × $\mu$ than their counterpart from non-G12 countries with the same number of Twitter mentions ($\beta_4 \times \mu$) = (0.021 × $\mu$). In order for a paper from a non-G12 country to reach the same number of citations as its US counterpart, our model suggests that it needs to have at least $\beta_3/\beta_4$ = 0.326/0.021 = 15.524 mentions to equalize the effect of the US. Such a level of Twitter visibility for a non-G12 paper is extremely rare (Fig 2), not to mention that papers from non-G12 countries are also less likely to appear in Q1 journals and to be published as OA (Table 3).

Topics appear to be an equalizing factor, which could potentially narrow the disparities in the impacts of communication research from US and non-US regions. Papers covering such topics have higher Twitter mentions and more citations. However, the equalizing effect of the "social media topic" has its limitations. As indicated by the interaction effects between topic

and country on citations: when both papers cover the "social media topic", the one from the US ($\beta_3 + \beta_6 + \beta_{14}$ = 0.326 + 0.794 + 0.225 = 1.345) would receive significantly more citations than the other from a non-G12 country ($\beta_6$ = 0.794). This finding supplements other studies' [40, 41] observation that certain topics are cited more frequently. However, this study found that US papers in these topics are cited even more.

As discussed in prior studies, OA's effects on publication impacts vary between disciplines and journals [71, 72]. Our study, which exclusively looks at communication studies, shows that OA does help papers to receive more citations. Although OA is practiced more by papers published by authors from industrial countries, the effect of OA on citations seems to be equalizing. The interaction effect of OA with countries of origin is not large enough to be significant. The interaction effect of OA and countries on Twitter mentions is even antagonistic for US papers. This suggests OA's potential as an equalizing force, which helps to balance the influence disparity between scholarship from developed and less developed countries in communication studies. However, OA was practiced in only 9% of papers from non-G12 countries. We therefore suggest that publishing papers as OA should be promoted universally and made easier particularly for researchers from underprivileged regions. Ideally, OA should be based on a completely free (both *gratis* and *libre*) "Platinum OA" model, like in the *International Journal of Communication* and *Computational Communication Research*. However, the reality is different: most commercial publishers promoting the so-called "Gold OA" charge article-processing fee (APC), which is not always waived for researchers from underprivileged countries. We recommend that journals, communication journals included, should make it easier for scholars from less developed countries to receive the APC waiver. For example, the $400 APC of *Social Media + Society* is waived for researchers from low income countries on a case-by-case basis, and PLOS provides a full waiver or a discount of the APC according to the GDP of the author's country.

## Conclusion

We conceptualize Twitter-worthiness as factors that make certain content more likely to be shared on Twitter than others. Based on prior literature on science communication and news value theory, we hypothesized that four factors are associated with the Twitter mentions of a scientific publication: topic, geographical origin (i.e. based in the US and G11 countries), publication in a high impact journal and with OA. We evaluated the effects of these factors by studying the Twitter mentions of 32187 papers published in communication journals between 2007 and 2018. Additionally, this study investigated to what extent Twitter visibility is associated with paper citations. Statistical analyses results have shown that all four factors mentioned before are independently associated with papers' visibility on Twitter, which also leads to more peer citations.

Findings suggest that Twitter mentions are associated with a higher citation impact of communication researchers' work, but the association is stronger for some than others. For instance, for papers by researchers located in less developed countries, the association between Twitter and citation is weaker. As mentioned earlier, regional disparities in impacts between its research communities are prominent in the field of communication studies. The possible benefit of Twitter to future citations of a paper has the potential to reinforce the disparity between the privileged researchers in Industrialized countries and the underprivileged researchers in the Global South; and, in turn, further enhances the cumulative advantage of the privileged cohort. Another key finding from this study is the equalizing potential of OA. We found that OA papers from researchers in non-G12 countries, in comparison to their counterparts in the US, have more Twitter mentions.

As compared to earlier research of social media's influence over academic outputs' impacts, this study has methodological advances. First, the current study includes a wider coverage of both journals and variables than prior studies on similar topics. More importantly, this study utilized the Bayesian approach for more reliable estimation of effect. Despite this paper's exclusive focus on Twitter, future research could apply similar methods to investigate how Twitter's impacts on scholarly communication might be echoed or offset by other social media platforms, such as Facebook or those with stronger regional foci (e.g. WeChat and VKontakte). This type of investigation could be particularly important in regions wherein Twitter does not play the dominant role in scholarly communication.

This study has several limitations. First, some factors that might be associated with Twitter mentions and citations are not considered in this study, such as the author's gender and ethnicity [15, 67]. We excluded these two factors for pragmatic reasons, as these two variables cannot be directly inferred from our available data. Manual coding is required to annotate such information [15, 67]. However, considering the size of our dataset, this process is not very feasible. Given the gender and ethnical disparities in academia, future development of the concept of Twitter-worthiness should take these two factors into consideration.

Second, not all existing communication journals are considered in this study. Although our coverage is likely more extensive than any previous study [15, 39–41, 67], there are still important omissions. Due to data limitations, the *International Journal of Communication*, a flagship journal of the field, was excluded. All journals not indexed by the *Web of Science* were also excluded. Some of these journals are of note (e.g. *First Monday* or many non-English journals such as *Publizistik*). These omissions are inevitable for all bibliometric studies relying on third party data sources such as Web of Science. It is also difficult to anticipate how these omissions could impact our findings.

Third, the data used in this study is cross-sectional and count-based, with both the Twitter mentions and citations obtained in 2019. Although the cross-sectional dataset is easier to analyze, it presents three problems. The first problem is that our dataset doesn't contain papers concerning current hot research topics that have been found to affect $v$, such as COVID-19 [73]. The second problem is that such a dataset is not suitable for time series analysis such as vector autoregression to determine the temporal precedence of events. Similarly, the analysis cannot reflect the interplay between temporal changes in Twitter users' behaviors and the communication science field. Since establishing the temporal precedence is the essential step to determine causality, all the findings obtained in this study are purely associational. Future studies of this kind should obtain time series data by following up included papers to obtain time-stamped Twitter mentions and citations regularly. The third problem is that because the count-based data does not contain information about "who cited whom" or "who mentioned whom", it is also not suitable for social network analysis. Therefore, the current analysis provides no insight into the potential role played by social network phenomena, such as homophily and core-periphery structure. Future studies of this kind should collect and analyze data that allows social network analysis.

Even with these limitations, our approach is more vigorous than previous studies, as it analyzes the most comprehensive collection of journals and takes into account more variables than previous studies on similar topics.

## Acknowledgments

The author would like to thank Professor Adrian Rauchfleisch (National University of Taiwan) for his comments on an earlier version of this paper, and Professor Eszter Hargittai (University

of Zurich) for her comment which inspired us to include Open access as one of the factors to test.

## Author Contributions

**Conceptualization:** Chung-hong Chan, Jing Zeng, Mike S. Schäfer.

**Data curation:** Chung-hong Chan.

**Investigation:** Chung-hong Chan, Jing Zeng, Mike S. Schäfer.

**Methodology:** Chung-hong Chan, Jing Zeng.

**Project administration:** Chung-hong Chan, Mike S. Schäfer.

**Resources:** Mike S. Schäfer.

**Software:** Chung-hong Chan.

**Supervision:** Mike S. Schäfer.

**Validation:** Jing Zeng.

**Writing – original draft:** Chung-hong Chan, Jing Zeng, Mike S. Schäfer.

**Writing – review & editing:** Chung-hong Chan, Jing Zeng, Mike S. Schäfer.

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
