## [Decision Letter · Decision Letter 0]

11 Jan 2022

PONE-D-21-23079Whose research benefits more from Twitter? On Twitter-worthiness of communication research and its role in reinforcing disparities of the fieldPLOS ONE

Dear Dr. Chan,

Thank you for submitting your manuscript to PLOS ONE. After careful consideration, we feel that it has merit but does not fully meet PLOS ONE’s publication criteria as it currently stands. Therefore, we invite you to submit a revised version of the manuscript that addresses the points raised during the review process.

Please address all the comments from the reviewers before submitting a revised version. The comments are quite detailed and will help the authors to have a better manuscript. 

We look forward to receiving your revised manuscript.

Kind regards,

Ronaldo Menezes

Academic Editor

PLOS ONE

Journal Requirements:

The name of the colleague or the details of the professional service that edited your manuscrip

Additional Editor Comments:

This work deals with an important issue regarding the usefulness of Twitter Data. The paper is sound and well written, however the reviewers have pointed out to several issues that need to be addressed before publication.

Reviewers' comments:

Reviewer's Responses to Questions

**Comments to the Author**

1. Is the manuscript technically sound, and do the data support the conclusions?

Reviewer #1: Partly

Reviewer #2: Partly

2. Has the statistical analysis been performed appropriately and rigorously? 

Reviewer #1: Yes

Reviewer #2: Yes

3. Have the authors made all data underlying the findings in their manuscript fully available?

Reviewer #1: Yes

Reviewer #2: Yes

4. Is the manuscript presented in an intelligible fashion and written in standard English?

Reviewer #1: Yes

Reviewer #2: Yes

5. Review Comments to the Author

Reviewer #1: This paper analyses the relation among Twitter mentions of communication research papers with four factors: geographical, research topic, high impact journal, and being open access. Then, the paper analyses the relationships of Twitter mentions with the number of citations and how this is corrected by the other factors. In general, the paper is sounding, and it makes important conclusions about the use of social media to promote scientific work with some open access recommendations. It has a good literature review and very good motivation about why their hypothesis and questions are worth studying. However, I consider the methods used for the analysis could improve their complexity to check better the hypotheses and support the claims made in the discussion. I have some major concerns and minor changes in the manuscript before being considered for publication.

Major comments:

1. Your hypothesis and results are interesting, but I think an analysis of them framed by the penetration of Twitter in the studied countries is important. The results regarding the importance of using Twitter can be driven by the use of Twitter in the country of affiliation. Then, a further discussion of other ways of tackling the "Mattew effect" in other countries can promote papers through the most used social media in each country.

2. Following the previous comment and your analysis of the already existing reward system in Academia before the use of Twitter, I encourage you to analyze if citations in the studied journals have changed trends due to the use of Twitter. Is it possible to establish that Twitter disturbed the previous growth of citations when countries/journals started to use it? How much has been accelerated due to the use of Twitter, and in which year the disruption started?

3. In addition, due to you have temporal data is there a way to measure reinforcing loops? Papers gaining citations then gain mentions and the same way around?

4. You mention several studies about the use of Twitter in other disciplines. How can other fields be compared with this? Is the use of Twitter more important in the Communication field because it is studying Twitter itself? Academicians using Twitter are those mentioning and citing studies about Twitter?

5. You have a central research question and then five hypotheses, and then an RQ1. I think the RQ1 and the main research question could be reframed and enclose all the hypotheses more precisely.

6. The way you divided the countries makes sense, but I think giant countries in production should also be analyzed in and outside the "global-south" work ... paper of developing countries.

7. For those Q1 journals that you analyzed, have you checked if the growth in citations is as expected or if the growth has been higher due to the Twitter mentions?

8. Your conclusion about the core/periphery structure of the communication scholarship is not supported by the analysis done in this article. Core/periphery structure is well studied under the network science approach, and such analysis is not done in the paper. For making this claim, it would be fantastic to find data for building the network of citations/collaborations of communication papers in this study and check the core/periphery position of the papers and its relation with the mentions in Twitter. It would be an incredible analysis. In case you want to pursue that analysis, it's worth reading this paper: "A clarified typology of core-periphery structure in networks."

Minor comments:

1. The abstract has some typos, e.g. visibile, disparties and singular/plural errors, e.g. "reinforce" should be "reinforces". Please, double-check.

2. Try to use the same name for the four factors or dimensions studied because the name change can be confusing. For example, publication outlets turn into journals and open access?

3. In the section: "Data collection & Operationalizations of variables", in lines 233 and 234, you mention 5 journals, but in the previous sentence, you mention 21, so it is confusing which are those 5.

4. The equations (1), (2) and (3) refer to two models that were explained very early in that section which makes it confusing what the models are. I recommend you to write explicitly the variables of each model: output and inputs.

5. Please, be consistent with the writing of Twitter-worthiness. Sometimes you write twitterworthiness (Line 266).

6. I recommend writing the words of acronyms the first time you use them to avoid confusion among readers. For example, in line 282: IQR.

7. When presenting the tables of the multilevel bayesian models, I would recommend having a name for each model because lines 296 and 313 introduce them similarly, and it isn't very clear.

8. For tables 2 and 3, it will be worth having a column with the values of the exponents to avoid confusion. Would you please double-check the values for the H2? Based on the table, I think some values got confused. For example, the results shown in lines 341-343 are not easy to follow.

9. Line 306 says QA instead of OA.

10. Regarding the conclusions about non-G12 countries, explaining how the values were calculated is important to support the claims in lines 344-349.

11. The font size of Figures 1 and 2 makes it difficult to read; it could be reduced. Please, write in the figure caption that n is the number of mentions to make it self-explanatory.

Reviewer #2: See the attached pdf xxxxxxxxxxxxxxxxxxxxxxxxxxxxxxxxxxxxxxxxxxxxxxxxxxxxxxxxxxxxxxxxxxxxxxxxxxxxxxxxxxxxxxxxxxxxxxxxxxxxxxxxxxxxxxxxxxxxxxxxxxxxxxxxxxxxxxxxxxxxxxxxxxxxxxxxxxxxxxxxxxxxxxxxxxxxxxxxxxx

6. PLOS authors have the option to publish the peer review history of their article (what does this mean?). If published, this will include your full peer review and any attached files.

Reviewer #1: **Yes: **Ana Maria Jaramillo

Reviewer #2: No

---

## [Author Response · Author response to Decision Letter 0]

25 Mar 2022

Dear editor(s) and reviewers,

We want to thank you for giving us the opportunity to revise our paper. We appreciate the two reviewers’ in-depth and constructive feedback. In the revised manuscript, we have incorporated comments from both reviewers. By following the reviewers’ suggestions, we believe that the clarity and discussion of our study have been improved a lot. We have conducted robustness checks recommended by the reviewers, and the result is located in the online repository (https://osf.io/hpva2/). The English grammar and spelling have been checked by a native English speaker. 

In the following section, we respond to each reviewer’s comments by explaining how and where changes were made. Reviewers’ comments are numbered according to their original order in the decision letter. 

Best regards,

Authors of manuscript PONE-D-21-23079

Reviewer One 

R1.1.Your hypothesis and results are interesting, but I think an analysis of them framed by the penetration of Twitter in the studied countries is important. The results regarding the importance of using Twitter can be driven by the use of Twitter in the country of affiliation. Then, a further discussion of other ways of tackling the "Mattew effect" in other countries can promote papers through the most used social media in each country.

Response: We appreciate this comment, and a similar point is mentioned by Reviewer Two (See R2.10). As we have now elaborated on Page 6, our operationalization of geographical origin factors in also the disparities in popularity of Twitter. We have now pointed out that industrialized countries also provided some of the largest user communities during the early days of Twitter. For instance, independent studies such as Hawelka et al. (2014) suggest that in 2012, roughly 30% of Twitter users were from the US alone. We also agree with the reviewer’s comment about the potential impact of other social media to tackle the Mathew effects. Although we cannot directly examine such association, we have now acknowledged that future research could apply our methods to look into how Twitter’s impacts on scholarly communication might be echoed or offset by other social media, such as Facebook or platforms with more regional foci (e.g. WeChat in China VKontakte in Russia). We have now argued that it is particularly important to conduct such research in regions wherein Twitter does not play the dominant role in scholarly communication.

R1.2. Following the previous comment and your analysis of the already existing reward system in Academia before the use of Twitter, I encourage you to analyze if citations in the studied journals have changed trends due to the use of Twitter. Is it possible to establish that Twitter disturbed the previous growth of citations when countries/journals started to use it? How much has been accelerated due to the use of Twitter, and in which year the disruption started?

R1.3. In addition, due to you have temporal data is there a way to measure reinforcing loops? Papers gaining citations then gain mentions and the same way around?

R1.7. For those Q1 journals that you analyzed, have you checked if the growth in citations is as expected or if the growth has been higher

due to the Twitter mentions?

Response: We would like to respond to these 3 comments together, because all of them concern the possibility of temporal analysis. In our discussion of limitations (Page 13, Lines 492-507), we have now explained why our cross-sectional and count-based dataset is not suitable for time series analysis. Both the number of citations and the number of Twitter mentions are cross-sectional data obtained in 2019. With the current data, it is not possible to do time series analysis, such as Vector Autoregression to determine reinforcing loops. However, we do see the value of such temporal analysis. We have now acknowledged this as a weakness of the dataset and pointed out that future studies of this kind should obtain time series data by obtaining time-stamped Twitter mentions and citations of included papers.

R1.4. You mention several studies about the use of Twitter in other disciplines. How can other fields be compared with this? Is the use of Twitter more important in the Communication field because it is studying Twitter itself? Academicians using Twitter are those mentioning and citing studies about Twitter?

Response: In the revised introduction section we have now more explicitly pointed out that (Lines 48-53) (1) Twitter is important for scholarly communication across disciplines, as informed by existing literature. So we do not argue that ‘Twitter is more important in the Communication field’; (2) at the same time, we have now pointed out that issues related to equity are prominent and it has become widely reflected upon from within the scholarly community of communication studies. This makes empirical results presented in the paper particularly relevant, (3) issues discussed in the paper should also be studied in other disciplines and our paper provides a methodological instrument for future research. 

R1.5. You have a central research question and then five hypotheses, and then an RQ1. I think the RQ1 and the main research question could be reframed and enclose all the hypotheses more precisely.

Response: We have changed our previous ‘central research questions’ into two clear analytical questions, and the previous ‘RQ1’ has been included as RQ3 ( Lines 36-31, 203-205) 

R1.6. The way you divided the countries makes sense, but I think giant countries in production should also be analyzed in and outside the "global-south" work ... paper of developing countries.

Response: We appreciate this comment. As geographical origin is the integral part of our conceptualization of Twitter-worthiness, we think it would be more appropriate to analyze all papers from all geographical regions altogether. Having said so, we consider the reviewer’s point as a robustness check in which only the US papers are considered. This robustness check is located in the online repository (https://osf.io/hpva2/).

R1.8. Your conclusion about the core/periphery structure of the communication scholarship is not supported by the analysis done in this article. Core/periphery structure is well studied under the network science approach, and such analysis is not done in the paper. For making this claim, it would be fantastic to find data for building the network of citations/collaborations of communication papers in this study and check the core/periphery position of the papers and its relation with the mentions in Twitter. It would be an incredible analysis. In case you want to pursue that analysis, it's worth reading this paper: "A clarified typology of core-periphery structure in networks."

Response: We would like to thank the reviewer for pointing this out. The “core/periphery” structure was mentioned because it is a well-known phenomenon in communication science (e.g. Dememter). But the reviewer is right that the current study does not provide any direct evidence (e.g. from social network analysis) to support it. Therefore, we decided not to mention it in the text.

R1.9. The abstract has some typos, e.g. visibile, disparties and singular/plural errors, e.g. "reinforce" should be "reinforces". Please, double-check.

Response: We have cleared all typos and grammatical errors in the abstract.

R1.10. Try to use the same name for the four factors or dimensions studied because the name change can be confusing. For example, publication outlets turn into journals and open access?

Response: We thank the reviewer for pointing this out. In the revised version, we have made it consistent that the two dimensions are High Impact Journals and Open Access (OA).

R1.11. In the section: "Data collection & Operationalizations of variables", in lines 233 and 234, you mention 5 journals, but in the previous sentence, you mention 21, so it is confusing which are those 5.

Response: There are actually 21 journals classified as Q1. The mentioning of ‘5’ was a mistake of failing to delete a footnote from a previous version of the paper. This mistake is now corrected.

R1.12. The equations (1), (2) and (3) refer to two models that were explained very early in that section which makes it confusing what the models are. I recommend you to write explicitly the variables of each model: output and inputs.

R1.15. When presenting the tables of the multilevel bayesian models, I would recommend having a name for each model because lines 296 and 313 introduce them similarly, and it isn't very clear.

Response: We have now revised the equations to spell out all terms, and we have also made sure that all terms are consistently named.

R1.13. Please, be consistent with the writing of Twitter-worthiness. Sometimes you write twitterworthiness (Line 266).

Response: We have checked these terms throughout the document to make sure that the spelling is consistent. 

R1.14. I recommend writing the words of acronyms the first time you use them to avoid confusion among readers. For example, in line 282: IQR.

Response: We have removed the reference to the mentioned acronym.

R1.16. For tables 2 and 3, it will be worth having a column with the values of the exponents to avoid confusion. Would you please double-check the values for the H2? Based on the table, I think some values got confused. For example, the results shown in lines 341-343 are not easy to follow.

Response: For the mentioned tables, we have added a column of exponents and we have double-checked the values for H2.

R1.17. Line 306 says QA instead of OA.

Response: The typo has been corrected in the manuscript.

R1.18. Regarding the conclusions about non-G12 countries, explaining how the values were calculated is important to support the claims in lines 344-349.

Response: We have made it clearer that the value is calculated by dividing the regression coefficient of US by the regression coefficient of mu.

R1.19. The font size of Figures 1 and 2 makes it difficult to read; it could be reduced. Please, write in the figure caption that n is the number of mentions to make it self-explanatory.

Response: We have reduced the font size of Figure 1 and 2 to make it easier to read. We have also added a caption to make it clearer that n denotes the number of articles.

Reviewer Two

R2.1. Introduction -“Social media might in turn become an enabler of the Matthew effect”. Explain specifically what is meant by “Matthew effect” and how social media can enable it.

Response: We replaced the mention of "Mattew effect" with "rich-gets-richer effect" and explained how social media can enable it: privileged researchers can benefit from social media but underprivileged researchers cannot (Lines 18-19).

R2.2 “The central question of this study is: Whose research benefits Twitter more? And then we ask: Can these benefits translate into actual citations?” Is this supposed to be “Whose research benefits from Twitter more”? In general the introduction is quite brief and could go into more detail about why this topic is important.

Response: We have revised the research question and extended the introduction by discussing more about the relevance and importance of the study. Please also see our response to R1.1.

R2.3. Data collection & Operationalizations of variables -“The authors’ countries of origin were extracted” How? Presumably not manually.

Response: We have now made it clearer that the information on “Geographical origin of a paper” was based on the address of the paper’s corresponding author. We extracted this information from the metadata provided by WoS (Lines 232-235). To avoid confusion, we have also made the description of this variable consistent, i.e. describing it as the original of a paper, rather than the origin of the author.

R2.4. “Using a topic modeling technique” Can you please give more information about this technique i.e. apart from using a dictionary, what is the algorithm?

Response: On Page 6 we have expanded the elaboration of how a keyATM model works (Lines 260-269).

R2.5 “because there are many social factors associated with citation behaviors.” For example?

Response: We added an example of these social factors from the literature: number of coauthors (Lines 307-309). 

R2.6. Statistical Analysis - This section is very succinct and could use much more elaboration. For example there appear to be 4 independent variables, which are a mixture of categorical (high impact, open access, geographic) and continuous (Topic membership θ), as predictors of citation/mention rate, but the number of variables is written as some unknown k and different symbols (G11, OA etc.) are used later in the paper where ‘x’ is used in this section.

“For our RQ1, we add interaction terms between factors determining twitterworthiness and region to all models”

Equations 1 and 3 are all rather trivial manipulations of each other and yet the model

described in the quote above is not given.

The model used to produce table 3 does not appear to be described and should be.

Response: In light of this comment and comment R1.12 mentioned above, we now have spelled out the two equations entirely on Page 8.

R2.7. Results - “We observe a very long-tail distribution of Twitter mentions.” Please include this figure if you are going to mention it.

Response: As the long-tail distribution of Twitter mentions is not important for the entire, we have decided to omit that therefore no figure is provided.

R2.8. “As three is the upper limit of the IQR for Twitter mentions, we use it in the 282 following two figures to define papers with high Twitter mentions” The upper limit is 4 in the G11 column for the mentions value.

Response: We have made it clearer that three is the upper limit of the interquartile range for $\\mu$ of all publications.

R2.9. Figure 2 - Does the population of a country play a role in citation/mention behaviour? Many social media analyses find a bias towards mentioning other users who are geographically closer (e.g. Han, S.Y., Tsou, M.H. and Clarke, K.C., 2018. Revisiting the death of geography in the era of Big Data: The friction of distance in cyberspace and real space. International Journal of Digital Earth, 11(5), pp.451-469).

Response: We would like to thank the reviewer for this interesting suggestion. However, due to the limitation of our dataset, we cannot pursue this analysis. The kind of relationship suggested by the reviewer requires directional data (who cites whom and who mentions whom), but, unfortunately, we don’t have such data. We have addressed this as a weakness in the discussion and suggested that future studies should collect such data to enable the analysis suggested by the reviewer. (please also see our reply to R1.2 above)

R2.10. The popularity of Twitter in a country also must play a huge factor in this analysis? E.g. in eastern Europe or China (i.e. much of the non-G12 category) Twitter is not very popular. Indeed, Twitter is officially blocked in China. Perhaps some other social media network is used there to disseminate research findings.

Response: We agree with the reviewer that the popularity of Twitter should also be a factor to consider. We have addressed this comment, please see our response to R1.1 from above.

R2.11. Table 3 - Are all of the values statistically significant? e.g. OA*G11 , OA*US

Response: Not all of the values are “statistically significant” in the traditional frequentist sense. The two interaction terms have a 95% HDI crossing zero. We have mentioned in the discussion that the two interaction effects are not large enough to be significant in the previous and the current version of the paper. [line 398]

R2.12. Discussion - Twitter has not been consistently popular since 2007 (https://trends.google.com/trends/explore?date=all&q=Twitter) and each country will have its own trajectory for Twitter use. The model does not account for this and it would be useful to see either robustness checks (e.g. a subset analysis of a shorter period when Twitter use is approximately stable) or at least some discussion of this issue.

Response: We agree with the reviewer that our model does not account for the global differences in the popularity of Twitter. We took the suggestion by the reviewer and conducted a robustness check with only the data from 2011 onwards. As reflected by the Google trend data, the global interest in Twitter is more even since 2011. The check suggests that our analysis is robust against this alternative explanation. We have provided this robustness check in the Online repository (https://osf.io/hpva2/).

R2.13. There is highly likely to be interplay between 1. Relative popularity of Twitter in US/G11 nations (measured by something like users per capita) 2. Number of communications studies researchers 3. Preference for mentioning/citing researchers geographically and culturally closer to the author. This warrants some discussion.

Response: We appreciate the reviewer’s suggestion, but due to the limitation of our data we cannot directly investigate the interplay mentioned above. Because the interplays suggested by the reviewers cannot be analyzed using our current cross-sectional count-based data, we addressed this as a limitation of the study (see R1.2 and R2.9). 

R2.14. Conclusions - “Additionally, this study investigated to what extent Twitter visibility improves paper Citations” This needs clarification, where was the direction of causality established? A fit of y = mx+c (cf. equation 3) only establishes correlation between x and y.

Response: We would like to thank the reviewer for pointing this out. Indeed the current analysis has no say in any causality. We addressed this point by 1) making it clear that this study can’t establish any causality claim (see our reply to R1.2 above), and 2) removing any rhetoric that could be interpreted as causality, e.g. “improves paper citations”, with rhetoric that conveys associations.

R2.15. General: A scan of the whole MS for English/grammar is necessary e.g. “such as colonialism influence” -> “such as the influence of colonialism”.

Response: We have corrected these grammatical errors. 

R2.16. I don’t doubt the findings of the paper and the methods are appropriate, however 1. A more thorough description of those methods is required 2. More robustness checks of the results are needed and 3. Recognition of the study’s limitations, especially with respect to the unknown effect of inhomogeneous Twitter data.

Response: We would like to thank the reviewer for endorsing our methods. For the three mentioned points, we addressed with the following:

We have a more thorough description of the methods (R2.3, R2.4)

We have conducted the suggested robustness checks (R2.12)

We have addressed the limitations of the study (R2.9, R2.13, R2.14)

---

## [Decision Letter · Decision Letter 1]

11 Oct 2022

PONE-D-21-23079R1Whose research benefits more from Twitter? On Twitter-worthiness of communication research and its role in reinforcing disparities of the fieldPLOS ONE

Dear Dr. Chan,

Thank you for submitting your manuscript to PLOS ONE. After careful consideration, we feel that it has merit but does not fully meet PLOS ONE’s publication criteria as it currently stands. Therefore, we invite you to submit a revised version of the manuscript that addresses the points raised during the review process.

We look forward to receiving your revised manuscript.

Kind regards,

Avanti Dey, PhD

Staff Editor

PLOS ONE

Journal Requirements:

Additional Editor Comments:

Can you please address the minor concerns raised by the reviewers?

Reviewers' comments:

Reviewer's Responses to Questions

**Comments to the Author**

1. If the authors have adequately addressed your comments raised in a previous round of review and you feel that this manuscript is now acceptable for publication, you may indicate that here to bypass the “Comments to the Author” section, enter your conflict of interest statement in the “Confidential to Editor” section, and submit your "Accept" recommendation.

Reviewer #3: All comments have been addressed

Reviewer #4: All comments have been addressed

2. Is the manuscript technically sound, and do the data support the conclusions?

Reviewer #3: Yes

Reviewer #4: Partly

3. Has the statistical analysis been performed appropriately and rigorously? 

Reviewer #3: N/A

Reviewer #4: Yes

4. Have the authors made all data underlying the findings in their manuscript fully available?

Reviewer #3: No

Reviewer #4: Yes

5. Is the manuscript presented in an intelligible fashion and written in standard English?

Reviewer #3: Yes

Reviewer #4: Yes

6. Review Comments to the Author

Reviewer #3: The authors have addressed the reviewers' comments fairly. The previous version's tone was too strong to be published. I believe the present version is adequate. However, several topics must still be addressed in the discussions. Communication journals, for example, with non-Western offices are uncommon. As a result, there is a discrepancy among the sampled articles. Furthermore, papers dealing with specific concerns tend to receive more attention. You might want to look into the following works.

Park, H.C., Youn, J.H., & Park, H.W.(2018). Global mapping of scientific information exchange using altmetric data. Quality & Quantity. 53(2), 935–955.

Holmberg,K., & Park,H.W.(2018). An altmetric investigation of the online visibility of South Korea-based scientific journals, Scientometrics. 117(1), 603–613

Chong,M., & Park,H.W.(2021). COVID-19 in the Twitterverse, from epidemic to pandemic: information-sharing behavior and Twitter as an information carrier. Scientometrics.

Park, H.J., Biddix, J.P., & Park, H.W. (2021). Discussion, news information, and research sharing on social media at the onset of Covid-19. EL PROFESIONAL DE LA INFORMACIÓN,v.30, n.4, e300405. 1-15.

Reviewer #4: The findings are very interesting and relevant to the area of information metric studies and research evaluation, but there is an important lack of information since the point of view of methods: You do not explain the software, method or code used to do the mining of Twitter, in order to recover all Twitter mentions. Even if you used WoS data to obtain this variable, you must explain it.

7. PLOS authors have the option to publish the peer review history of their article (what does this mean?). If published, this will include your full peer review and any attached files.

Reviewer #3: No

Reviewer #4: **Yes: **Antonio Eleazar Serrano López

---

## [Author Response · Author response to Decision Letter 1]

24 Nov 2022

Dear editor and dear reviewers,

Thank you for your insightful feedback on our research paper “Whose research benefits more from Twitter? On Twitter-worthiness of communication research and its role in reinforcing disparities of the field.” (PONE-D-21-23079R1). Following the reviewers’ suggestions, we have revised the manuscript. We respond below through a point-by-point outline.

We hope that the revised manuscript will now be suited for publication in PLOS ONE. We are looking forward to hearing back from you. 

Sincerely,

The Authors

Reviewer #3

Comment 1.1 The authors have addressed the reviewers' comments fairly. The previous version's tone was too strong to be published. I believe the present version is adequate. However, several topics must still be addressed in the discussions. Communication journals, for example, with non-Western offices are uncommon. As a result, there is a discrepancy among the sampled articles. 

Response 1.1: We would like to thank the reviewer for acknowledging the improvement made in our previous revision, and for pointing out issues concerning the differences between journals (e.g. journals with or without non-Western offices). We fully agree with the point raised by the reviewer. We did model that in our Bayesian models with the varying intercept. In the revised manuscript we have strengthened this point in lines 302-5:

“There might be individual differences in µ and ν according to characteristics of journals, e.g. countries of origin [65]. By adding a varying intercept according to the j-th journal as uoj to model the individual differences in µ and ν, our two models are expressed as….”

Comment 1.2 Furthermore, papers dealing with specific concerns tend to receive more attention. 

Response 1.2: We agree with the reviewer that studies related to some concerns (e.g. Covid, as the papers suggested by the reviewer) may receive more social media engagements than others. Our paper has taken it into consideration and we modelled it as “Hot research topic” (θt). As shown in the revised manuscript (Lines 105-111), our conceptualization of twitter-worthiness has also incorporated this point. However, because our dataset was collected in 2019, the current analysis cannot capture new research concerns such as Covid. We have admitted this as a weakness in lines 470-3 and cited the important work by Chong & Park (2021):

“Third, the data used in this study is cross-sectional and count-based, with both the Twitter mentions and citations obtained in 2019. Although the cross-sectional dataset is easier to analyze, it presents three problems. The first problem is that our dataset doesn’t contain papers concerning current hot research topics which have been found to affect ν, such as COVID-19 [73].”

Comment 1.3 You might want to look into the following works.

Park, H.C., Youn, J.H., & Park, H.W.(2018). Global mapping of scientific information exchange using altmetric data. Quality & Quantity. 53(2), 935–955.

Holmberg,K., & Park,H.W.(2018). An altmetric investigation of the online visibility of South Korea-based scientific journals, Scientometrics. 117(1), 603–613

Chong,M., & Park,H.W.(2021). COVID-19 in the Twitterverse, from epidemic to pandemic: information-sharing behavior and Twitter as an information carrier. Scientometrics.

Park, H.J., Biddix, J.P., & Park, H.W. (2021). Discussion, news information, and research sharing on social media at the onset of Covid-19. EL PROFESIONAL DE LA INFORMACIÓN,v.30, n.4, e300405. 1-15.

Response 1.3 We thank the reviewer for recommending the above references. We find them helpful and have incorporated them in the revised manuscript.

Reviewer #4

Comment 2.1 The findings are very interesting and relevant to the area of information metric studies and research evaluation, but there is an important lack of information since the point of view of methods: You do not explain the software, method or code used to do the mining of Twitter, in order to recover all Twitter mentions. Even if you used WoS data to obtain this variable, you must explain it.

Response 2.1 We would like to thank the reviewer for finding this study interesting and relevant. We agree with the reviewer that we could have provided more clarification on how Twitter-related data was collected. We have now added information about the software and method of our Twitter data in lines 207-211:

‘Altmetric, according to its website, “tracks Twitter attention in real-time via an API ... collect[s] tweets, retweets, and quoted tweets that contain a direct link to a scholarly output.” [57] Employing the Altmetric API and the R package rAltmetric [58], we extracted the mention counts of all 32187 articles in June 2019, with the DOIs.’

---

## [Editor Report · Decision Letter 2]

28 Nov 2022

Whose research benefits more from Twitter? On Twitter-worthiness of communication research and its role in reinforcing disparities of the field

PONE-D-21-23079R2

Dear Dr. Chan,

We’re pleased to inform you that your manuscript has been judged scientifically suitable for publication and will be formally accepted for publication once it meets all outstanding technical requirements.

Kind regards,

Pablo Dorta-González, Ph.D.

Academic Editor

PLOS ONE
---

## [Editor Report · Acceptance letter]

1 Dec 2022

PONE-D-21-23079R2 

Whose research benefits more from Twitter? On Twitter-worthiness of communication research and its role in reinforcing disparities of the field 

Dear Dr. Chan:

I'm pleased to inform you that your manuscript has been deemed suitable for publication in PLOS ONE. Congratulations! Your manuscript is now with our production department. 

Kind regards, 

on behalf of

Mr. Pablo Dorta-González 

Academic Editor

PLOS ONE